# Integrating Polygenic Scores into Multifactorial Breast Cancer Risk Assessment: Insights from the First Year of Clinical Implementation in Western Austria

**DOI:** 10.3390/cancers17152472

**Published:** 2025-07-26

**Authors:** Lukas Forer, Gunda Schwaninger, Kathrin Taxer, Florian Schnitzer, Daniel Egle, Johannes Zschocke, Simon Schnaiter

**Affiliations:** 1Insititute of Genetic Epidemiology, Department of Genetics, Molecular and Clinical Pharmacology, Medical University of Innsbruck, 6020 Innsbruck, Austria; lukas.forer@i-med.ac.at (L.F.); florian.schnitzer@i-med.ac.at (F.S.); 2Institute of Human Genetics, Department of Genetics, Molecular and Clinical Pharmacology, Medical University of Innsbruck, 6020 Innsbruck, Austria; gunda.schwaninger@i-med.ac.at (G.S.); johannes.zschocke@i-med.ac.at (J.Z.); 3Department of Gynecology, Brust Gesundheit Zentrum Tirol, Medical University of Innsbruck, 6020 Innsbruck, Austria

**Keywords:** breast neoplasms, risk assessment, genetic counseling, precision medicine, predictive value of tests

## Abstract

Genetic cancer risk assessment in clinical practice often focuses on single cancer risk genes without considering the level of variation in the genetic background. In this study, we employed polygenic score (PGS) analysis in the setting of multifactorial risk assessment (MFRA), combining a person’s genetic information, family history, personal health data and lifestyle factors to estimate their risk of getting breast cancer, in a cohort of 13 women with monogenic moderate-risk variants in Western Austria. The personal risk was either estimated for contralateral breast cancer in women diagnosed with breast cancer (*n* = 8) or for primary breast cancer in women without a cancer diagnosis (*n* = 5). Integrating PGS information sometimes led to substantial changes (more than 10% change in 5 of 13 cases) in the predicted inherited breast cancer risks and influenced surgical management decisions in 5 cases (38%). Our findings support the use of PGS in future breast cancer genetic assessments with MFRA to improve patient diagnosis and care.

## 1. Introduction

Breast cancer is the most commonly diagnosed malignancy among women worldwide, including Austria. Despite substantial advances in treatment and early detection, particularly through screening programs, it remains a leading cause of cancer-related mortality. Its etiology is multifactorial and involves both environmental and inherited factors, as well as chance.

Genetic testing for breast cancer susceptibility, particularly in women with a strong family history of breast and ovarian cancer, is well established for the determination of inherited breast cancer risk. With advances in technology and decreasing sequencing costs, the inclusion criteria for testing are becoming less stringent, and the scope of genetic testing has expanded from high-risk genes, such as *BRCA1*, *BRCA2,* and *PALB2*, to encompass more moderate-risk genes, such as *CHEK2*, *ATM*, *RAD51C*, *RAD51D*, and *BARD1* [1]. Clinical management for individuals who are heterozygous for likely pathogenic and pathogenic variants (summarized as PVs in this manuscript) in high-risk genes is already well established, often providing the option of risk-reducing surgery in addition to intensified surveillance.

Testing genes that confer only a moderate increase in breast cancer risk may sometimes complicate the situation. PVs in such moderate-risk genes lead to challenging decisions regarding optimal preventive and screening measures. The introduction of the CanRisk tool (BOADICEA) enables a multifactorial risk assessment (MFRA) for breast cancer and represents a major step forward in personalized risk predictions [2,3,4]. The MFRA has shown that the risk associated with PVs in moderate penetrance genes varies significantly, from below the population average to risk levels that are comparable to the risk conferred by high-risk PVs [2,5,6]. Additionally, the inclusion of individual data allows for a more precise quantification of personalized breast cancer risk.

The MFRA is based on epidemiologic models that integrate clinical factors (e.g., breast density), lifestyle factors (e.g., contraception, smoking, and alcohol consumption), family history, and genetic background in the form of a polygenic score (PGS). The PGS quantifies the cumulative impact of many low-penetrance genetic variants in the human genome and thereby enhances risk stratification [7]. Tools like CanRisk [4], or comparable algorithms [8,9], facilitate the integration of the MFRA to support counseling by genetic counselors and clinical geneticists and clinical management by oncologists. However, this approach may also present ethical (e.g., equitable access across diverse populations) and technical/computational challenges (e.g., GDPR-compliant genotype imputation [10,11]). It also may be difficult because of resource limitations (e.g., genetic testing and counseling capacities) [4,12] and with regard to integration into existing workflows [13].

### 1.1. MFRA to Support Genetic Counseling

Predicting cancer risk is a prerequisite for an informed approach to genetic counseling and balancing the personal benefits versus adverse effects of surveillance, as well as risk-reducing measures like preventive therapy and risk-reducing surgery. In addition to clinical management, as some of these factors are modifiable, they can be incorporated into counseling strategies to help women at risk of breast cancer adopt risk-reducing behaviors, such as lowering BMI, reducing alcohol consumption, and limiting the use of hormone replacement therapy [2]. These algorithms are increasingly used in complex clinical cases in Austria to inform decision-making practices during genetic counseling and are on the edge of entering standard clinical practice [14]. Therefore, they are playing a growing role in the informed decision-making of patients during the genetic counseling process.

### 1.2. The Need for the MFRA Is Most Obvious in Patients with PVs in Breast Cancer Risk Susceptibility Genes of Moderate Penetrance

In Austria, genetic testing for hereditary breast and ovarian cancer is funded by the public healthcare system for individuals with an increased risk based on their personal clinical and family history. Initially, only high-penetrance genes such as *BRCA1*, *BRCA2*, and *PALB2* (approx. 50–80% lifetime risk for breast cancer) [15] were included. At our center, the panel was expanded in 2022 to identify PVs in moderate-penetrance genes, such as *CHEK2*, *ATM*, *RAD51C*, *RAD51D*, and *BARD1* (approx. 20–40% lifetime risk for breast cancer), as well as the so-called syndromic genes, *PTEN*, *STK11,* and *TP53,* which are associated with other specific manifestations [1]. This expansion led to an increase in individuals with PVs in breast cancer risk genes. By leading to an increased number of identified carriers of PVs in moderate-risk genes, this expansion also presents challenges in the counseling and clinical management of these patients [16]. Therefore, a rising need for more precise risk prediction strategies evolved.

Typically, women with an anticipated elevated breast cancer risk undergo genetic testing and enter intensified surveillance programs based on their personal and family histories. Upon detection of a PV in a high-penetrance breast cancer risk gene, additional measures, such as risk-reducing surgery, are offered. These measures are not usually offered to patients with moderate-risk PVs, meaning that the identification of such a variant does not necessarily result in additional clinical action, as the women are usually already enrolled in intensified surveillance. Hence, testing for moderate risk genes without including additional risk modifiers has little clinical consequence. It has been shown that the personalization of risk prediction via MFRA allows for the reclassification of about 50% of individuals with heterozygous PVs in moderate-risk breast cancer genes, from either moderate risk to high risk or moderate risk to low risk, with implications for counseling and clinical management [2]. Accordingly, the introduction of MFRA for individuals with heterozygous PVs in moderate-risk breast cancer genes is a necessary consequence of testing for variants in moderate-risk breast cancer genes.

### 1.3. Implementation of Polygenic Scores and MFRA in Breast Cancer Prevention and the Clinical Management of Breast Cancer

In addition to PVs in high- or moderate-risk genes that significantly increase the likelihood of developing breast cancer (although to a varying extent), genome-wide association studies (GWASs) have identified multiple other genetic variants associated with breast cancer susceptibility [17]. While each of these variants poses a minimal individual risk, their cumulative effect, quantified in a polygenic score (PGS), can be significant [18], allowing for targeted screening and preventative strategies to enhance breast cancer prevention and survival [19]. The PGS alone does not usually lead to an adaptation of counseling and clinical management [7], but it may be used to substantially enhance the predictive power of the MFRA. Thus, we integrated the BCAC313 score [7] into the MFRA using the CanRisk tool (CanRisk v3.0.0) to improve personalized breast cancer prevention and/or the personalized clinical management of breast cancer.

### 1.4. Preemptive Considerations for Maximal Patient Safety and Acceptance

A key challenge in implementing MFRA and PGS within an established clinical setting is the uncertainty it creates for medical geneticists, genetic counselors, oncologists, and most importantly, patients. To gain experience with the new risk assessment approach, a restrictive reporting policy was adopted, in which MFRA results that could lead to a reduced intensity of surveillance were reported but not quantified. This conservative approach enabled the clinical team to build familiarity with the model while at the same time minimizing potential patient harm.

In this article, we present our approach and first experiences from one year of MFRAs, including PGSs, into the clinical management of women with moderate-penetrance risk variants in a regional center in Western Austria.

## 2. Methods

This is a real-world evidence study.

### 2.1. Polygenic Score Integration into Multifactorial Assessment

The PGS implemented into our MFRA framework is the BCAC313, which is listed as PGS00004 (Release: 14 October 2019) in the PGS Catalogue [20]. This score comprises 313 genetic variants and was developed using data from >150,000 individuals of European ancestry across 69 distinct cohorts. Exact details about this score can be found in Mavaddat et al. [7].

In brief, the PGS analysis for each individual patient consisted of the following steps (Figure 1):
Genotyping across a representative set of genome-wide variants using microarray;Performing rigorous quality control of the genotype data;Imputation of missing genotypes using the 1000 Genomes Reference Panel;Estimation of the ancestry of the patient with a principal component analysis (PCA);Computation of the raw PGS;Standardization of the score with a local reference population (z-score).

### 2.2. Genotyping, Quality Control, and Imputation

Genotyping was performed using the Infinium™ Global Screening Array with Cytogenetics-24 v1.0 (Illumina, San Diego, CA, USA), as described previously [21]. Quality control of genotyping array data was carried out with the pre-imputation-qc pipeline v1.2.0 (https://github.com/genepi/pre-imputation-qc, accessed on 28 May 2025) using appropriate reference data for the GSAMD-24v3-0-EA_20034606_A1.b37 chip (https://www.chg.ox.ac.uk/~wrayner/strand/, accessed on 28 May 2025). The QC pipeline excludes (i) indels, (ii) non-autosomal SNPs, (iii) SNPs with low minor allele frequency or with extreme Hardy–Weinberg equilibrium deviations, and (iv) SNPs with call rates below <0.9. In addition, strand orientation is corrected, alleles are harmonized, and only SNPs with known reference alleles are retained. Also, samples with call rates <0.5 (or with at least one 20MB region with a call rate of <0.5) were excluded. No patient sample had to be excluded due to insufficient technical data quality.

Imputation of the QCed genotype data is performed using a local instance of imputationserver2 (https://github.com/genepi/imputationserver2, accessed on 28 May 2025) [10] and the 1000 Genomes Phase 3 reference panel [11]. To ensure stable imputation, each patient’s sample was processed together with all samples from the local reference cohort (described below), which were also submitted to the same QC procedures as the patient samples.

### 2.3. Ancestry Estimation

Ancestry estimation for each patient sample was achieved through a principal component analysis (PCA) of genotype data using LASER [22]. The ancestry reference panel consisted of 938 individuals from the Human Genome Diversity Project [23]. Ancestry naming was adjusted to reflect the super populations recommended by Morales et al. [24]. Each patient was classified based on similarity (e.g., distance on the PCA space) to the reference samples using a K-Nearest Neighbor (K-NN) algorithm. Further details about the applied ancestry estimation procedure are published in Forer et al. [25].

Samples with uncertain ancestry estimation or non-European ancestry were excluded, since the applied PGS was developed exclusively with individuals of European ancestry, and the PGS accuracy is reduced in individuals whose ancestry does not match or is not represented in the PGS training cohort [26].

### 2.4. Polygenic Score Calculation

The raw polygenic score for BCAC313 (PGS000004) was calculated with the imputed data using the integrated Imputation Server PGS functionality [25]. In brief, the PGS was calculated by summing the products of variant-specific weights and corresponding genotype dosages across all PGS variants. The raw PGS for each patient was normalized using the European ancestry samples of the local reference cohort (*n* = 1235). Each batch of patient samples was processed together with the local reference cohort, allowing PGS values to be calculated for all samples. This provided a reference distribution, which was used to calibrate the PGS specifically to the local population.

The normalized z-score for each patient ($z_{i}$) was computed by subtracting the raw score (*PGS_i_*) from the mean and dividing it by the standard deviation (*sd*) of the PGS values in the local reference cohort (*PGS_ref_*) as follows:
$$z_{i}=\frac{{PGS}_{i}-mean({PGS}_{ref})}{sd({PGS}_{ref})}$$

As suggested by Mavaddat et al. [27], the PGS calculation had to be adjusted for patients carrying the *CHEK2* variant c.1100del because this mutation is integrated into the MFRA via gene-panel testing, and two variants of PGS000004 (22:29203724:C:T and 22:29551872:A:G) are correlated with this genomic variant. In such cases, PGS integration would lead to an overestimation of risk, since the BOADICEA framework generally assumes that PGS and gene panel genotypes contribute to breast cancer risk independently [27]. Therefore, the PGS was computed without the two *CHEK2* correlated variants (22:29203724:C:T and 22:29551872:A:G) for patients carrying the *CHEK2*-PV c.1100del. In these cases, the PGSs for the reference samples were also calculated with the reduced variant set to ensure valid z-score normalization for this adjusted variant set.

## 3. Results

### 3.1. Patients

During the study period, from January to December 2024, 457 women were tested for hereditary breast and ovarian cancer according to Austrian national guidelines [28]. In this cohort, 17 women tested positive for a PV in a moderate-risk gene and consented to a full MFRA, including the calculation of the breast cancer PGS BCAC313 (CONSORT diagram; see Appendix A). One patient was of non-European descent and was therefore excluded, as the PGS results would have been of limited validity [29]. One patient was excluded due to insufficient clinical data. Two patients, one with a *CHEK2*-PV and one with an *ATM*-PV, had bilateral breast cancer, rendering the MFRA for breast cancer obsolete. Consequently, 13 heterozygotes with a PV in a moderate-risk gene received a full MFRA, including PGS. Five of these individuals presented for cascade testing, i.e., targeted testing for a PV previously identified in a biological relative, and were cancer-free, while eight individuals had unilateral breast cancer. Individual details are shown in Table 1.

Of note, four women with low-penetrance *CHEK2*-PV c.470T>C [30] were included in our cohort; one of them carried the variant in combination with a *BARD1*-PV. The *CHEK2* variant c.470T>C is included in the BCAC313 PGS [7] as a variant/SNP with a high effect size and consequently was not included as a standalone PV in the MFRA.

### 3.2. Cumulative Effect of MFRA and PGS Testing

A PGS was generated for 15 patients. In patients with breast cancer or bilateral breast cancer, we observed significantly (*p* = 0.016) elevated PGS z-scores, with a median of 1.525, which is in the 10th decile compared to the hypothetical population mean of 0.0 (Figure 2A). In comparison, in healthy PV-heterozygotes, we observed z-scores with a mean of 0.55 (in the eighth decile), which was not significantly different from the hypothetical population mean. In both groups, the range was broad, with z-scores ranging from −1.5 to 2.4 for breast cancer patients and from −0.8 to 2.9 for healthy PV-heterozygotes, reflecting the variability in the individual genetic backgrounds.

The elevated mean PGS z-score was independent of the gene in which the PV was found (Figure 2B).

Next, we applied the MFRA model to the whole cohort, beginning with the population-level risk estimates. At the population level, the individual risk varies due to variable age and age of diagnosis/variant detection in heterozygotes. As expected, since our cohort was pre-selected for a positive breast cancer family history and individuals with moderate-penetrance variants, the inclusion of PVs, as well as family and clinical data, predicted a substantially increased future breast cancer risk (Figure 2C,D): it increased from 12% (age-matched population risk) to 23% in patients with breast cancer (risk for contralateral breast cancer) and from 10% (age matched population risk) to an average risk of 24% in healthy heterozygotes (risk for breast cancer). The addition of the PGS led to a further increase in risk to 26% for breast cancer patients and to 33% for healthy heterozygotes (Figure 2C,D). As anticipated, there was a strong deviation of the individual risk from the average risk in our cohort, as indicated by the error bars in Figure 2C,D.

### 3.3. Impact of Individualized Risk Prediction

Next, the individual impact of the MFRA on the 13 patients who received full MFRAs was analyzed. The individual scores for each patient are depicted graphically and numerically in Figure 3A,B.

-For three breast cancer patients (aged 46–62 years), the MFRA had little or no impact. These patients already had a low population risk of 5.9–8.5% for contralateral breast cancer due to their comparably higher age. Following the MFRA, their individual risk increased from 8.2% to 12.2%.-Three breast cancer patients (aged 49–59 years) experienced a moderate increase in risk to 24.9–29.5% following the MFRA, compared to a population risk of 8.4% to 14.2%. One of these patients had an exceptionally low PGS that reduced the individual risk by more than 10%.-Two relatively young breast cancer patients (aged 37 and 41 years) experienced a marked increase in individual risk to 38.4% and 59.9% following the MFRA, compared to population risks of 19.7% and 23.3%, respectively.-Five of the patients with breast cancer decided to undergo bilateral mastectomies. However, the relative influence of the risk predicted by the MFRA, including PGS, compared to other clinical and non-clinical factors could not be quantified.-For two healthy heterozygotes in our cohort of patients (aged 41 and 61 years), the MFRA had a minor or moderate impact, with calculated risks of 17.1% and 20.7%, respectively (age-matched population risks of 7.3% and 11.4%, respectively), for breast cancer.-Three healthy heterozygotes (aged 27–56 years) experienced a moderate to strong increase in risk to 30.3–58.2% following the MFRA, with a population risk between 8.2 and 12%.

Notably, even in our rather small cohort, the PGS alone modified the individual breast cancer risk or contralateral breast cancer risk by more than 10% in four out of thirteen patients and by nearly 10% in two additional patients (Figure 3B).

## 4. Discussion

The goal of implementing the MFRA is to accurately predict the individual risk of breast cancer by integrating genetic, clinical, and lifestyle data. In this study, we assessed the impact of the MFRA, including PGS, in women with elevated breast cancer risk at our center. We observed high PGSs in most breast cancer patients who carried PVs in moderate-risk genes but less marked effects in healthy heterozygotes. The consideration of MFRA, including PGS, led to substantially increased risk estimates, particularly for contralateral breast cancer in women with previous breast cancer.

It was expected that the PGS would be increased in breast cancer patients who carried PVs in moderate-risk genes, but not in healthy heterozygotes, as a cohort of breast cancer patients is, of course, pre-selected for an increased breast cancer risk in comparison to the general population. The same is true of the overall increase in risk exhibited by the MFRA model in contralateral breast cancer. It is of key importance, however, that the individual risk for contralateral breast cancer, even in a cohort of heterozygotes of PVs in moderate-penetrance genes, varies greatly, ranging from nearly the same as the general population to an individual risk as high as 59.9%. Meanwhile, a generally increased risk in healthy individuals from breast cancer families carrying a moderate penetrance gene PV was evident. However, the inclusion of other risk factors, particularly age, breast density, and PGS, leads to a significant variation in individual risk, ranging from 17.1% to 58.2%. Hence, our observations underline the importance of applying the MFRA, including PGS, to comprehensively understand individual risk in the context of multiple impact factors. Our cohort was too small to allow for a reliable assessment of the relative impact of individual risk factors in the MFRA. However, our observations are consistent with previously published data [2] indicating that very high or very low levels of PGS, breast density, and family history exert the strongest relative influence of the assessed risk factors—each comparable to, and in some extreme cases, exceeding the impact of PVs in moderate-risk genes.

Our study highlighted that combining multiple individual risk factors enhances the personalized risk assessment for healthy individuals and breast cancer patients with heterozygous moderate-penetrance risk variants, compared to relying solely on population-average risk estimates. Advanced risk models, such as the BOADICEA algorithm used in the CanRisk tool, integrate monogenic data, polygenic scores, family history, lifestyle factors, and imaging data. This approach is particularly valuable when counseling individuals with variants in moderate-penetrance breast cancer genes like *CHEK2*, *ATM*, *RAD51C*, *RAD51D,* and *BARD1*, since the relative effect of risk factors is expected to be higher than in individuals with high-penetrance PVs. By providing advanced personalized risk stratification, such models enable more precise risk predictions and more informative cascade screenings of healthy relatives. Furthermore, tailored recommendations for surveillance, prevention, and management strategies can be offered to breast cancer patients and healthy carriers. The tool presents risk estimates in an accessible format, thereby empowering women to understand their options for lifestyle modifications, screening, or preventive measures, such as risk-reducing surgery. Ultimately, the comprehensive approach in the MFRA supports the shared decision-making process that is central to effective genetic counseling and patient-centered care.

In daily clinical practice, the potential consequences of the MFRA in the management of individuals heterozygous for moderate-penetrance PVs are apparent in individualized risk stratification, guiding decisions on screening frequency, early intervention, and even chemoprevention. They provide an additional layer of precision for communicating risk to patients and supporting informed decision-making. Closing the gaps in unknown risk situations is a much-awaited development for clinicians. However, key challenges remain. The major challenge is that most PGSs are based on data from populations of European ancestry, limiting their validity in more diverse populations [31]. Recently adapted versions of BCAC313 based on data from people of non-European ancestry and corresponding updates in CanRisk are promising, suggesting that this problem can be alleviated in the near future [32]. The implementation of the adopted PGS in our bioinformatics pipeline is in preparation. A lack of standardized thresholds and clinical guidelines complicates the use of PGS in routine practice. While the full MFRA model, including the PGS, allows for classification into defined risk classes (e.g., NICE classification) [33] by oncologists, a major concern remains: the lack of conclusive studies on optimal management strategies for high-risk patients beyond traditional high-risk PV carriers [15]. Effective strategies to convey predictive genetic risk without causing undue anxiety are in development [13]. Yet, there is a need for more solid long-term data on the clinical impact of PGS. The collection of real-world evidence in a controlled patient collective is one way of collecting the required data to generate an actionable threshold.

## 5. Limitations of This Study

The outcome of this study is limited by the small sample size and the heterogeneity of the cohort. The general relevance of PGS and MFRA has been shown before in large, well controlled cohort studies [5,6]. This study was designed to demonstrate the feasibility and necessity of applying the PGS and MFRA for individual risk prediction in routine care. Accordingly, while a larger sample size would have been welcome, the small cohort was sufficient to draw valuable conclusions.

## 6. Conclusions

In conclusion, our study supports the notion that using the MFRA, including the PGS, enhances the precision of breast cancer risk prediction in carriers of moderate-penetrance gene variants. Despite the small cohort size, our findings illustrate that this approach can meaningfully improve individual risk estimates and influence clinical decision-making. The observed risk variability in carriers of PVs in moderate-risk genes demonstrates the strong impact of other factors, such as the PGS, breast density, and family history. This finding highlights the need to move past categorizations of genetic risk based on PVs in single genes and encourages the adoption of more nuanced and personalized models. As the MFRA becomes increasingly accessible through tools like CanRisk, it holds the potential to transform genetic counselling and support preventive strategies in breast cancer care. Validation in other population groups, along with the development of clear clinical guidelines and standardized computational workflows, is needed to unlock the maximum potential of the MFRA.

This study highlights that testing for PVs in moderate-risk genes should be followed by personalized risk assessment using the MFRA, including the PGS, as the cumulative individualized risk can differ substantially from the average risk associated with an isolated PV. By showing that genetic cancer risk extends beyond PVs in single genes and that this multifactorial risk can be efficiently assessed, this study aims to support the integration of the MFRA and PGS into guidelines and routine care within national health systems.

## Figures and Tables

**Figure 1 cancers-17-02472-f001:**
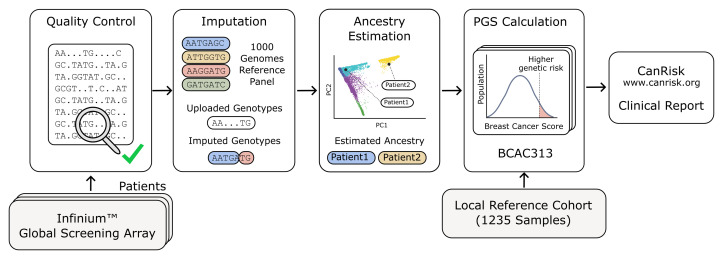
Flowchart showing bioinformatics processing of raw data to calculate the PGS.

**Figure 2 cancers-17-02472-f002:**
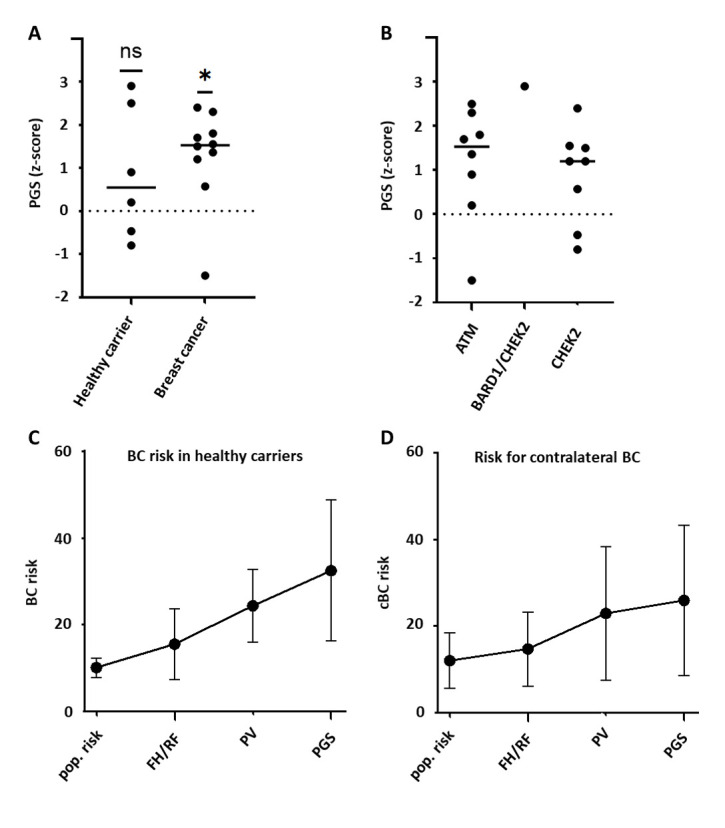
Cumulative effect of MFRA and PGS testing in the cohort. (**A**) PGS z-score is significantly increased in BC patients, but not in healthy heterozygotes (Wilcoxon signed rank test). (**B**) PGS z-score is not different between *ATM* and *CHEK2* variant heterozygotes. (**C**,**D**) The predicted average risk for breast cancer in healthy women (**C**), as well as the risk for contralateral breast cancer in BC patients (**D**), increased with the application of the MFRA. Abbreviations: BD—breast cancer, cBC—contralateral breast cancer, Pop. risk—population risk, FH—family history, RF—risk factor, PV—pathogenic variant, PGS—polygenic score, ns—not significant, *—significant.

**Figure 3 cancers-17-02472-f003:**
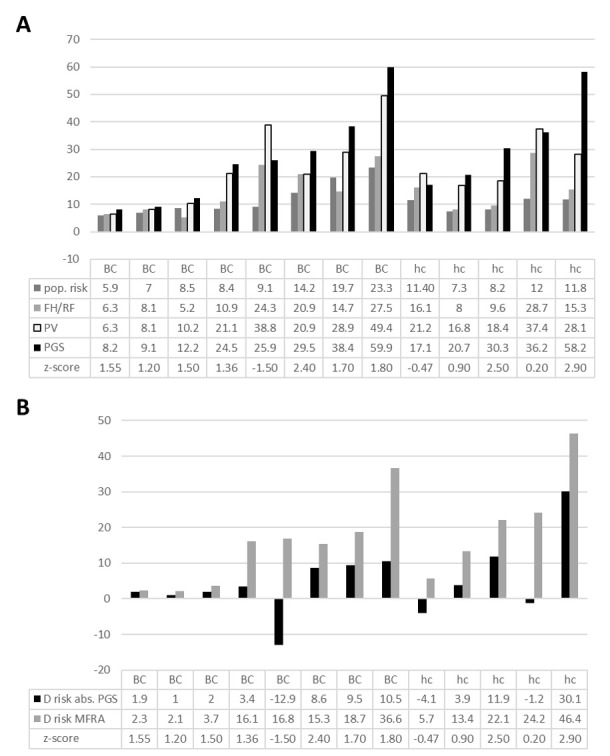
Personalized risk prediction. (**A**) Absolute risk prediction for individual patients/healthy heterozygotes; (**B**) absolute contribution of PGS (black bars) to the overall risk (grey bars). Abbreviations: BC—breast cancer patient, hc—healthy heterozygote, PGS—polygenic score, FH—family history, RF—risk factor, PV—pathogenic variant, MFRA—multifactorial risk prediction.

**Table 1 cancers-17-02472-t001:** Patient details.

Patient ID	Breast Cancer (Age of Diagnosis)	Age at Testing	Gene	Variant	ACMG Classification
1	BC (42), cBC (57)	58	*ATM*	c.3802del	C5
2	BC (49), cBC (57)	57	*CHEK2*	c.(908+1_909-1)_(1095+1_1096-1)del	C5
3	BC (35)	37	*ATM*	c.(4236+1_4237-1)_(4436+1_4437-1)del	C5
4	BC (41)	41	*ATM*	c.7674del	C5
5	BC (46)	62	*CHEK2*	c.792+2T>C	C4
6	BC (49)	49	*CHEK2*	c.470T>C	risk factor
7	BC (57)	57	*ATM*	c.3137T>C	C4
8	BC (59)	59	*ATM*	c.5644C>T	C5
9	BC (62)	63	*CHEK2*	c.470T>C	risk factor
10	BC (60)	62	*CHEK2*	c.470T>C	risk factor
11	p	41	*CHEK2*	c.1100del	C5
			*PMS2*	c.1831dup	C5
12	p	35	*BARD1*	c.1725_1903+1208del	C5
			*CHEK2*	c.470T>C	risk factor
13	p	56	*ATM*	c.3802del	C5
14	p	27	*ATM*	c.4148C>A	C5
15	p	60	*ATM*	c.8584G>T	C5

Abbreviations: BC—breast cancer, cBC—contralateral breast cancer, p—predictive testing, C4—likely pathogenic, C5—pathogenic variant

## Data Availability

The raw data supporting the conclusions of this article will be made available by S.S. upon reasonable request.

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
