# Peer review of "Integrating Polygenic Scores into Multifactorial Breast Cancer Risk Assessment: Insights from the First Year of Clinical Implementation in Western Austria"

_cancers, 2025, doi:10.3390/cancers17152472_

Round 1
Reviewer 1 Report
Comments and Suggestions for Authors
This manuscript titled "Integrating Polygenic Scores into Multifactorial Breast Cancer Risk Assessment: Insights from the First Year of Clinical Implementation in Western Austria" presents a timely and clinically relevant study evaluating the integration of polygenic risk scores (PGS) within multifactorial risk assessment (MFRA) for breast cancer. The authors report real-world data from a regional center in Austria, demonstrating how the inclusion of PGS can refine risk stratification in carriers of moderate-penetrance variants and influence clinical decision-making, including surgical management. The study is well-structured, methodologically sound, and offers valuable insights into the translational application of PGS in personalized cancer risk counseling. I commend the authors for their efforts and provide the following comments for revisions to improve clarity, context, and impact.
Comments
Abstract
- Line 22–23: "Substantial changes of the predicted inherited breast cancer risks and influenced surgical management..." – the term "substantial" is subjective. Consider quantifying the change here (e.g., “≥10% change in 5 of 13 cases”).
Introduction
- Line 67–69: Sentence beginning with “The inclusion of genes…” is long and convoluted. Consider breaking into two sentences for clarity.
- Line 85: Add more recent citations discussing real-world clinical implementation of MFRA tools.
- Line 99: The transition between discussions on Austria’s national testing policy and MFRA could be smoother. Suggest connecting policy and implementation outcomes more explicitly.
Methods
- Line 148: It’s helpful to cite the specific version of PGS000004 and provide the release date or data freeze.
- Lines 162–173: Impressive QC pipeline description; however, you could clarify how missing data affected the number of included patients.
- Line 204–213: The CHEK2 adjustment is thoughtful. However, provide evidence or a reference supporting this decision’s impact on prediction accuracy.
Results
- Line 223: Clarify “cascade testing” briefly for readers outside clinical genetics.
- Line 229–233: The inclusion of CHEK2 c.470T>C as a risk factor rather than a PV could use stronger justification or a reference.
- Lines 237–240: Since the cohort is small, report median and range for z-scores and add confidence intervals where applicable.
- Line 275: The statement “Five of the patients... underwent bilateral mastectomy” is impactful. Please clarify whether this was directly influenced by MFRA results.
Discussion
- Line 306–308: Consider briefly discussing which risk factors had the largest individual impact beyond PGS (e.g., breast density, family history).
- Line 333–335: The mention of European ancestry limitations is critical. Consider expanding this to include a discussion on steps needed for inclusion of non-European populations.
- Line 336–339: Excellent point on clinical ambiguity. Recommend suggesting pathways for developing actionable thresholds.
Conclusion
- Line 347: "Overly simplistic categorizations" could be better defined (e.g., binary risk classification).
- Suggest reinforcing how this study informs future guidelines or integration into national health systems.
Author Response
|
Response to Reviewer 1 Comments
|
||
|
1. Summary |
|
|
|
Thank you very much for taking the time to review this manuscript. Please find the detailed responses below and the corresponding revisions/corrections highlighted/in track changes in the re-submitted files.
|
||
|
2. Questions for General Evaluation |
Reviewer’s Evaluation |
Response and Revisions |
|
Does the introduction provide sufficient background and include all relevant references? |
Yes |
|
|
Is the research design appropriate? |
Yes |
|
|
Are the methods adequately described? |
Can be improved |
Please find an improved description of methods in the revised manuscript. |
|
Are the results clearly presented? |
Can be improved |
Please find an improved description of results in the revised manuscript. |
|
Are the conclusions supported by the results? |
Yes |
|
|
Are all figures and tables clear and well-presented?
|
Yes |
|
Comments and Suggestions for Authors
This manuscript titled "Integrating Polygenic Scores into Multifactorial Breast Cancer Risk Assessment: Insights from the First Year of Clinical Implementation in Western Austria" presents a timely and clinically relevant study evaluating the integration of polygenic risk scores (PGS) within multifactorial risk assessment (MFRA) for breast cancer. The authors report real-world data from a regional center in Austria, demonstrating how the inclusion of PGS can refine risk stratification in carriers of moderate-penetrance variants and influence clinical decision-making, including surgical management. The study is well-structured, methodologically sound, and offers valuable insights into the translational application of PGS in personalized cancer risk counseling. I commend the authors for their efforts and provide the following comments for revisions to improve clarity, context, and impact.
Comments
Abstract
- Line 22–23: "Substantial changes of the predicted inherited breast cancer risks and influenced surgical management..." – the term "substantial" is subjective. Consider quantifying the change here (e.g., “≥10% change in 5 of 13 cases”).
Response 1: The sentence (now line 24-25) was modified to “Integrating PGS information sometimes led to substantial changes (more than 10% change in 5 out of 13 cases) of the predicted inherited breast…”
Introduction
- Line 67–69: Sentence beginning with “The inclusion of genes…” is long and convoluted. Consider breaking into two sentences for clarity.
Respose 1: The sentence (now line 70-71) was modified/split as follows: “Testing of genes that confer only a moderate increase of breast cancer risk may sometimes lead complicate the situation. PVs in such moderate risk genes lead to challenging deci-sions regarding optimal preventive and screening measures.”
- Line 85: Add more recent citations discussing real-world clinical implementation of MFRA tools.
Response 2: The citations Archer et al., 2020; PMID: 32142536 (cited before at a different position) and Archer et al., 2023; PMID: 37308304 (new) are now included at the position (now line 88).
- Line 99: The transition between discussions on Austria’s national testing policy and MFRA could be smoother. Suggest connecting policy and implementation outcomes more explicitly.
- Response 3: The paragraph (now line 110 – 115) was supplemented accordingly.
Methods
- Line 148: It’s helpful to cite the specific version of PGS000004 and provide the release date or data freeze.
Response 1: The release data/data freeze of the version used is now specified in the text (now line 155).
- Lines 162–173: Impressive QC pipeline description; however, you could clarify how missing data affected the number of included patients.
Response 2: We added the sentence “No patient sample had to be excluded due to insufficient technical data quality.” (now line 180)
- Line 204–213: The CHEK2 adjustment is thoughtful. However, provide evidence or a reference supporting this decision’s impact on prediction accuracy.
Response 3: We clarified in the text, that the adjustment has been suggested and validated in previous studies.
Results
- Line 223: Clarify “cascade testing” briefly for readers outside clinical genetics.
Response 1: The meaning of “cascade testing” has been clarified by modifying the sentence (now line 230) from “Five of these individuals presented for cascade testing and were cancer-free,…” to “Five of these individuals presented for cascade testing, i.e. targeted testing for a PV previously identified in a biological relative, and were cancer-free,…”
- Line 229–233: The inclusion of CHEK2 c.470T>C as a risk factor rather than a PV could use stronger justification or a reference.
Response 2: A citation was added and the text (now lines 236 to 239) was modified to clarify that this PV must not be considered as independent PV wen using the BCAC313.
- Lines 237–240: Since the cohort is small, report median and range for z-scores and add confidence intervals where applicable.
Response 3: Median and the range of z-scores were added. The authors would appreciate not to ad confidence intervals, since in this paragraph an observation, rather than a predictive statistical result, is reported. Actually, personalization of risk prediction assumes heterogeneous results. The paragraph (now 240 to 249) has been modified to make that clearer.
- Line 275: The statement “Five of the patients... underwent bilateral mastectomy” is impactful. Please clarify whether this was directly influenced by MFRA results.
Response 4: We are aware that this information would be very valuable. However, suitable measures to capture that information have not been taken. We clarified that in the text by adding a sentence (now Line 286-288).
Discussion
- Line 306–308: Consider briefly discussing which risk factors had the largest individual impact beyond PGS (e.g., breast density, family history).
Response 1: This is a very interesting, and underestimated question. Even though our data does not allow a generalized statement due to the sample size, we extended the discussion on that point (now lines 322-327).
- Line 333–335: The mention of European ancestry limitations is critical. Consider expanding this to include a discussion on steps needed for inclusion of non-European populations.
Response 2: The corresponding text (now lines 351-355) has been modified to clarify this point.
- Line 336–339: Excellent point on clinical ambiguity. Recommend suggesting pathways for developing actionable thresholds.
Response 3: The according paragraph (now line 357-364) has been modified accordingly.
Conclusion
- Line 347: "Overly simplistic categorizations" could be better defined (e.g., binary risk classification).
Response 1: The paragraph (now 377 to 379) was adopted, including requirements by another reviewer.
- Suggest reinforcing how this study informs future guidelines or integration into national health systems.
Response 2: A final paragraph on how this study informs future guidelines or integration into national health systems has been added (line 386 to 391).
Reviewer 2 Report
Comments and Suggestions for Authors
This is a well written manuscript with a novel approach to integrating polygenic risk scores into a multifactorial risk assessment.
There are suggestions for further clarification of sections within the manuscript.
Simple summary:
1- revise to include the cohort of 13 women included both women with a breast cancer diagnosis and healthy women ( no cancer diagnosis)
Abstract:
- Line 38- clarify what is meant by " in the majority of patients, MFRA including PGS increased....
- line 42- in 5 cases, the modified risk assessment contributed to surgical decision for prophylactic mastectomy.... please clarify the specific PV mutation status of these "5 cases"
Introduction:
- MFRA to support genetic counseling- this section should include what is meant by "genetic counseling" ? with a genetic counselor, in person, virtual or PCP/Oncologist
- Implementation of polygenic scores and MFRA in clinics
- please clarify what is meant by 'in clinics"- is this a Primary care setting or oncology/specialty clinics
- Increasing compliance by minimizing risk
- I do not understand the relevance of this paragraph. It fits better under a "limitation"
- Impact of individualized risk prediction
- this section is very difficult to read and interpret. I would strongly recommend putting this individual patient scores into a table
- Limitations
- please provide limitations of this manuscript
- very small sample size
- heterogenous cohort and not generalizable
- please provide limitations of this manuscript
- Conclusion
- please clarify in the conclusion that the small sample size makes it difficult to support the MFRA /PGS to enhance precision of breast cancer risk prediction in carriers
Author Response
|
Response to Reviewer 2 Comments
|
||
|
1. Summary |
|
|
|
Thank you very much for taking the time to review this manuscript. Please find the detailed responses below and the corresponding revisions/corrections highlighted/in track changes in the re-submitted files.
|
||
|
2. Questions for General Evaluation |
Reviewer’s Evaluation |
Response and Revisions |
|
Does the introduction provide sufficient background and include all relevant references? |
Yes |
|
|
Is the research design appropriate? |
Can be improved |
We aware of certain limitations of the study, e.g. sample size etc.. We are presenting data from patients in routine care during the year 2024 and such are limited by the number and characteristics of these patients. |
|
Are the methods adequately described? |
Can be improved |
Please find an improved description of methods in the revised manuscript. |
|
Are the results clearly presented? |
Can be improved |
Please find an improved description of results in the revised manuscript. |
|
Are the conclusions supported by the results? |
Can be improved |
Please find an improved description of methods in the revised manuscript
|
|
Are all figures and tables clear and well-presented?
|
Yes |
|
|
3. Point-by-point response to Comments and Suggestions for Authors
Simple summary: 1- revise to include the cohort of 13 women included both women with a breast cancer diagnosis and healthy women ( no cancer diagnosis). Response 1: The simple summary has been revised by the addition of the sentence: “… 13 women with monogenic moderate-risk variants in Western Austria. The personal risk was either estimated for contralateral breast cancer in women diagnosed with breast cancer (n=8) or for primary breast cancer in women without a cancer diagnosis (n=5).” Abstract:
Response 1: The phrase “In the majority of the patients, MFRA including PGS increased risk estimates for both contralateral breast cancer in patients with breast cancer and for primary breast cancer in healthy carriers, compared to the risk conferred by the moderate-penetrance pathogenic variant alone.” was adjusted as follows: “MFRA including PGS increased risk estimates for both contralateral breast cancer in 7 out of 8 patients with breast cancer and for primary breast cancer in 3 out of 5 healthy carriers, compared to the risk conferred by MFRA and the moderate-penetrance pathogenic variant alone.”
Response 2: The sentence “In 5 cases, the modified risk assessment contributed to surgical decision for prophylactic mastectomy....” was modified as follows: “In five cases, one with a CHEK2-PV and four with an ATM-PV, the modified risk assessment contributed to surgical decision for prophylactic mastectomy....” We fully appreciate the interest of the reviewer in more details about the clinical management. However, we are limited by the ethical approval in that context, since anonymization/pseudonymization might be infringed by description of individual PVs and clinical management. Introduction:
Response 1.1: The sentence “…facilitate the integration of MFRA to support personalized counseling and prevention strategies in a clinical setting.” was modified as follows: “…facilitate the integration of MFRA to support personalized counseling by genetic councilors and clinical geneticists and clinical management by oncologists.”
Response 2.1: The phrase “in clinics” was replaced by “in breast cancer prevention and clinical management of breast cancer” in the paragraphs headline. The last sentence in the paragraph was complemented by the clause “…to improve personalized breast banker prevention and/or personalized clinical management of breast cancer.” to improve comprehensibility.
Response 3.1: We are the first institution in Austria to apply MFRA and PGS in patient diagnosis in Austria. In this initial phase we wanted to minimize the risk for patients while gaining experience. At the same time we expected, and faced, hesitant acceptance by our partners for this new method. For the safety of the patients, and to overcome resentments, we developed a rather conservative preemptive strategy of application of MFRA and PGS. This paragraph was designed to briefly reflect this considerations. For a better understanding for the reader the headline was altered from “Increasing compliance by minimizing risk” to “Preemptive considerations for maximal patient safety and acceptance”
Response 4.1: We fully aware of the complexity of the paragraph. The individual scores are contained in the table in figure 3a and 3b. For an improved understanding for the reader, we included the sentence: “The individual scores for each patient are depicted graphically in Figure 3a and numeri-cally in the table below the graph in Figure 3a and 3b respectively.” The readability of the numbers in the table is improved in the high resolution full size image supplied along with the compiled manuscript.
Response 5.1.1 and 5.1.2: A limitations paragraph has been added.
Response 6.1: Even though the sample size is small, we see variability in the individual risk. Our goal was not the generalized analysis of a cohort, but to demonstrate that the individual risk is diverse. Such our data supports that MFRA/PGS to enhances precision of breast cancer risk prediction. We added modified the corresponding sentence in the conclusions. “The observed variability in risk in carriers of PVs in moderate risk genes demonstrates the strong impact of other factors such as PGS, breast density or family history. That high-lights the need to move past categorizations of genetic risk based on PVs in single genes and encourages the adoption of more nuanced and personalized models.”
|
||
Round 2
Reviewer 2 Report
Comments and Suggestions for Authors
The authors have appropriately addressed reviewer comments.
Please review for minor spelling errors... eg line 141... should be 'cancer" and not banker...